# Impact of Induction Immunosuppressants on T Lymphocyte Subsets after Kidney Transplantation: A Prospective Observational Study with Focus on Anti-Thymocyte Globulin and Basiliximab Induction Therapies

**DOI:** 10.3390/ijms241814288

**Published:** 2023-09-19

**Authors:** Hyung Duk Kim, Hyunjoo Bae, Sojeong Yun, Hanbi Lee, Sang Hun Eum, Chul Woo Yang, Eun-Jee Oh, Byung Ha Chung

**Affiliations:** 1Division of Nephrology, Department of Internal Medicine, Eunpyeong St. Mary’s Hospital, College of Medicine, The Catholic University of Korea, Seoul 03312, Republic of Korea; scamph@catholic.ac.kr; 2Department of Biomedical Science, Graduate School, The Catholic University of Korea, Seoul 06591, Republic of Korea; jaydcom8673@gmail.com (H.B.); dbsthwjd789@naver.com (S.Y.); 3Division of Nephrology, Department of Internal Medicine, Seoul St. Mary’s Hospital, College of Medicine, The Catholic University of Korea, Seoul 06591, Republic of Korea; hanbilee89@gmail.com (H.L.); yangch@catholic.ac.kr (C.W.Y.); 4Division of Nephrology, Department of Internal Medicine, Incheon St. Mary’s Hospital, College of Medicine, The Catholic University of Korea, Incheon 21431, Republic of Korea; trickyspot@gmail.com; 5Department of Laboratory Medicine, Seoul St. Mary’s Hospital, College of Medicine, The Catholic University of Korea, Seoul 06591, Republic of Korea; 6Research and Development Institute for In Vitro Diagnostic Medical Devices, The Catholic University of Korea, Seoul 06591, Republic of Korea

**Keywords:** anti-thymocyte globulin, basiliximab, kidney transplantation, T lymphocyte subsets

## Abstract

Induction immunosuppressive therapy for kidney transplant recipients (KTRs) primarily includes interleukin-2 receptor antagonists, such as basiliximab (BXM) or lymphocyte-depleting agents, and anti-thymocyte globulin (ATG). This study aimed to investigate their effects on T cell dynamics during the early post-transplantation period. This prospective observational study included 157 KTRs. Peripheral blood samples were collected from each patient within 5 days before and 4 and 12 weeks after transplantation. Flow cytometric analysis was performed to assess various T cell subsets whose changes were then analyzed. In the ATG group, CD4^+^ T cell expression decreased significantly compared with that in the BXM group. However, CD4^+^CD161^+^ and CD4^+^CD25^+^CD127low T cell expression levels increased significantly. In the CD8^+^ T cell subset, a decrease in CD8^+^CD28nullCD57^+^ and CD8^+^CCR7^+^ T cell expression was observed in the ATG group. However, among patients diagnosed with biopsy-proven acute rejection, T cell subset expression did not significantly differ relative to non-rejection cases. In conclusion, ATG induction therapy resulted in more pronounced changes in T lymphocyte subsets than BXM induction, with increased CD4^+^CD161^+^ and CD4^+^CD25^+^CD127low T cells and an early decrease in CD8^+^CD28nullCD57^+^ and CD8^+^CCR7^+^ T cells, some of which are associated with acute rejection.

## 1. Introduction

Induction therapy is an essential component of immunosuppressive treatment for kidney transplantation (KT) that modulates the T cell response immediately after allograft implantation, thereby reducing the risk of acute allograft rejection [1,2,3]. This also makes it possible to reduce the dose of maintenance immunosuppressants, such as calcineurin inhibitors and corticosteroids [1]. Widely used induction therapies include interleukin-2 receptor antagonists, such as basiliximab (BXM), and lymphocyte-depleting agents, such as anti-thymocyte globulin (ATG) [4,5]. One study revealed that from 2002 to 2017, 77% of kidney transplant recipients (KTRs) in Korea received BXM induction therapy, while 12% received ATG induction therapy [6].

Although both drugs target T cells, they exhibit significant differences in their mechanisms of action. BXM suppresses T cells by interfering with the action of interleukin-2 (IL-2) through selective binding to the α-subunit (CD25) of the IL-2 receptor on the surface of activated T lymphocytes [7]. Therefore, it can suppress T cell proliferation but does not deplete T cells. In contrast, ATG is a polyclonal antibody against thymocytes, purified from rabbits or horses, that can deplete CD4^+^ T, CD8^+^ T, CD20^+^ B, and CD16^+^/56^+^ NK cells in a dose-dependent manner [8,9]. These differences in T cell phenotypes, attributed to the choice of induction therapy, may have clinical implications. Notably, investigations comparing clinical outcomes following different induction therapies have consistently indicated a potential reduction in the risk of acute rejection with ATG induction treatment [10,11]. Nevertheless, a comprehensive understanding of the dynamics of T cell phenotypes based on induction therapy remains elusive.

In previous studies, we demonstrated a significant association between specific immune cell phenotypes in peripheral blood mononuclear cells, laboratory biomarkers, and the clinical and histological status of kidney allografts [12,13,14]. This highlighted that the decrease in naïve and central memory T cell counts, coupled with the increase in effector memory T cell counts, is associated with acute allograft rejection. Therefore, we hypothesized that different types of induction therapies might result in changes in the production of Th17, effector memory, or Treg cells. In the present study, we aimed to explore the potential disparities in immune cell phenotypes based on different induction therapies to elucidate the differential impacts of these therapeutic approaches on clinical outcomes. We focused on Th17 cell activation and Treg cell expression, which may contribute to allograft function during KT. To this end, we extensively investigated the proportions of lymphocyte subsets, including CD4^+^CD161^+^, CD4^+^CD28nullCD57^+^CD161^+^, and CD8^+^CD28nullCD57^+^ T cells, as they have been associated with Th17 activation. Additionally, we comprehensively assessed the regulatory T cell populations using CD25, CD127, CCR7, and CD45RA markers. Our primary objective was to identify and characterize the specific T cell phenotypes associated with ATG or BXM induction therapy during the early post-transplant period.

## 2. Results

### 2.1. Comparison of Baseline Characteristics According to Induction Therapy

A total of 157 patients participated in the study, with 62 and 95 receiving ATG and BXM treatments, respectively. In the ATG group, a higher proportion of patients were sensitized and showed positive panel-reactive antibody (PRA) or donor-specific human leukocyte antigen (HLA) antibody (DS-HLA) results. Additionally, both PRA classes I and II % were elevated in ATG-treated patients. The frequency of rituximab administration was higher in the ATG group than that in the control group. Although the number of HLA mismatches was slightly higher in the ATG group, this difference was not statistically significant. Notably, there were no significant differences in patient age, sex, or donor type between the two groups (Table 1).

### 2.2. Clinical Outcomes According to Induction Therapy

In the ATG group, 3 biopsy-proven acute rejections (BPAR) were reported in 62 patients. Among these cases, two were diagnosed with acute T cell-mediated rejection (ATCMR), and two were identified as having acute antibody-mediated rejection (AABMR). One patient experienced a mixed rejection episode, simultaneously presenting with both ATCMR and AABMR. The BXM group encountered 10 BPAR incidents among the 95 patients, all classified as ATCMR. The incidence of BPAR did not differ significantly between the two groups. The mean time for diagnosing rejection was 14.1 weeks after transplantation, with occurrences ranging between 1 and 7 months post transplantation. Throughout the study period, neither group experienced cytomegalovirus (CMV) infection. However, BK viremia was observed in 2 patients in the ATG group and 11 in the BXM group. Although BK viremia appeared higher in the BXM group, this difference was not statistically significant. Furthermore, two cases of graft failure were observed in the BXM group (Table 2).

### 2.3. Comparison of Changes in CD4^+^ T Cell Subsets according to Induction Therapy

To investigate the dynamics of CD4^+^ T cell subsets, we initially examined the expression of CD3, CD4, CD28, CD57, and CD161. We measured the cell count for each lymphocyte subset and expressed it as a percentage of the total lymphocytes. When comparing the differences in CD4^+^ T cell subsets according to induction therapy, CD4^+^ T cell expression did not differ between the two groups at baseline (Figure 1a). However, at 4 and 12 weeks after transplantation, the ATG group exhibited a significant decrease in CD4^+^ T cell expression, whereas the BXM group showed no difference from baseline (Figure 1d). Similarly, CD4^+^CD161^+^ T cell expression did not differ significantly between the two groups at baseline (Figure 1c). Nevertheless, both groups showed an increase at 4 weeks post transplantation, with the ATG group demonstrating a significantly higher increase. At 12 weeks, both groups showed a slight decrease, but the ATG group maintained higher levels than those at baseline, whereas the BXM group showed no difference (Figure 1f). No significant difference was observed in CD4^+^CD25^+^CD127low T cell expression between the two groups (Figure 1h). However, it was observed that CD4^+^CD25^+^CD127low T cell expression in the ATG group was significantly increased 12 weeks after transplantation compared with that at baseline (Figure 1k). In the BXM group, CD4^+^CD25^+^CD127low T cell expression 4 and 12 weeks after transplantation showed no difference from baseline levels (Table 3).

### 2.4. Comparison of Changes in CD8^+^ T Cell Subsets according to Induction Therapy

At 12 weeks after transplantation, a noticeable increase in CD8^+^ T cell expression was observed in the ATG group (Figure 1b). Although the BXM group also exhibited a significant increase in CD8^+^ T cell expression at week 12 relative to that at baseline, the levels remained significantly lower than those in the ATG group. CD8^+^CD28nullCD57^+^ T cells showed no significant changes after transplantation in the BXM group (Figure 1i). In contrast, the ATG group showed a significant decrease in CD8^+^CD28nullCD57^+^ T cell expression 4 weeks after transplantation compared with that at baseline. The decreased expression in the ATG group significantly increased 12 weeks after transplantation, reaching levels similar to those at baseline (Figure 1l). CD8^+^CCR7^+^ T cell expression was lower at 4 and 12 weeks after transplantation in the ATG group than that in the BXM group (Figure 2a). In the ATG group, a decrease in CD8^+^CCR7^+^ T cell expression was observed at both 4 and 12 weeks compared with that at baseline (Figure 2d). The BXM group showed a minor decrease compared with the ATG group but exhibited a significant decrease at 12 weeks relative to baseline levels. Both CD8^+^CCR7^+^CD45RA^-^ and CD8^+^CCR7^+^CD45RA^+^ T cell expressions showed a decreasing pattern at 12 weeks compared with that at baseline in both groups (Figure 2e,f). The expression rates were higher in the ATG group than in the BXM group (Figure 2b,c).

### 2.5. Changes in T Cell Subsets in Patients with BPAR

To evaluate changes in lymphocyte subsets associated with rejection, we analyzed the expression of T lymphocyte subsets in the 13 patients diagnosed with BPAR (Figure 3). At 12 weeks post transplantation, CD4^+^ T cell expression was significantly decreased compared with that at baseline, whereas the other subsets showed no significant changes over time. At 4 weeks after transplantation, CD4^+^CD161^+^ T cell expression slightly increased, whereas that of CD4^+^CD25^+^CD127low T cells decreased. CD8^+^CD28nullCD57^+^ T cell expression decreased slightly at 4 weeks and then increased again at 12 weeks. CD8^+^CCR7^+^ T cell expression decreased at 12 weeks; however, the difference was not statistically significant compared with the baseline.

## 3. Discussion

In this study, we investigated the effects of induction therapy on the expression of T cell subsets during the early post-transplantation period. We compared T cell subset changes following ATG and BXM induction therapy in KTRs. Our primary focus was the change in Th17 and Treg cells with ATG induction. ATG, a polyclonal antibody targeting thymocytes, is widely used as an induction immunosuppressive therapy in solid organ or bone marrow transplantation. It exerts its effects on various immune cells through diverse mechanisms [8,15]. For instance, it induces T cell and antigen-presenting cell depletion via complement-dependent lysis or antibody-dependent cellular cytotoxicity and triggers apoptosis in activated T cells. Furthermore, it maintains dendritic cells in an immature tolerogenic state while impeding the function of mature dendritic cells [8]. Additionally, it modulates cell surface molecules involved in cell adherence and trafficking, such as CXCR4 and CCR7 [16]. Unlike BXM, which specifically inhibits IL-2 receptor action, the use of ATG with its diverse mechanisms led us to anticipate significant alterations in subsets important in transplantation immunology, such as Th17 and Treg cells.

First, we analyzed the differences in lymphocyte subsets based on induction therapy to verify the preventive effects of ATG induction on acute rejection in highly sensitized patients—clinical outcomes according to induction therapy were compared. In addition, we examined the patients who experienced BPAR in detail to identify the lymphocyte subset expression related to rejection.

Regarding baseline characteristics, the two induction therapy groups exhibited no significant differences in the demographic variables. However, as expected, the ATG induction group comprised highly sensitized patients, resulting in significantly higher values of immunological parameters, such as PRA and DS-HLA, than those in the BXM group. The choice of induction therapy was determined according to our transplant clinic’s routine practice and the medical team’s judgment. Therefore, patients receiving ATG induction were expected to have a higher immunological risk than those receiving BXM induction, potentially leading to more rejections and graft failures. In our study, ATG induction could effectively prevent rejection and graft failure, showing non-inferior clinical outcomes compared with BXM induction (Section 2.2). Additionally, given that ATG generally possesses stronger immunosuppressive effects, we aimed to show that the frequency of infectious complications was not at concerning levels. The clinical outcomes depicted in Table 2 showed no significant differences between the two groups regarding BPAR and graft failure. Although not statistically significant, there was a tendency for lower BPAR, graft failure, and BK viremia frequencies in the ATG group compared with those in the BXM group. Although there was no significant difference in the incidence of BPAR, that in the BXM group (10.5%) was slightly higher than that in the ATG group (4.8%). Notably, when comparing ATCMR cases, while there was no statistically significant difference, the incidence rate differed between the two groups, with the BXM group showing a higher rate than that of the ATG group (10.5% vs. 3.2%). Moreover, two cases of AABMR occurred in the ATG group, possibly due to the higher proportion of highly sensitized patients in this group. The two groups had no significant difference in CMV infection or BK viremia incidence. However, graft failure was observed in two cases, both in the BXM group.

Some previous studies suggested that the proportion of CD4^+^CD161^+^ T cells, which represent Th17 cells, increased in patients with allograft injuries [17,18,19]. CD4^+^CD161^+^ T cells were also found to stimulate an immunological response by producing inflammatory cytokines such as IL-17 [17,18]. Loverre et al. reported that increased CD4^+^IL-17^+^ cell infiltration within renal tubules could be mediated by complement system activation and was found in KTRs diagnosed with acute antibody-mediated rejection [20]. However, in the present study, CD4^+^CD161^+^ T cell expression was increased in the ATG group, contrary to our initial expectations. This unexpected outcome could be attributed to the composition of the ATG group, which consisted of highly sensitized patients with a higher immunological risk. Moreover, the relatively short study duration of 12 weeks may have introduced confounding effects due to the impact of early ischemic-reperfusion injury of the graft. The increased CD4^+^CD161^+^ T cell expression at 4 weeks post-transplantation in both the ATG and BXM groups, compared with the baseline, suggests the potential influence of ischemic-reperfusion injury on CD4^+^CD161^+^ T cell expression during the early post-transplantation period.

Treg cells inhibit activated T cells and regulate immunity [21,22]. CD4^+^CD25^+^ and FOXP3^+^ Treg cell expression negatively correlates with acute rejection [23]. In the results of the Immune Development in Pediatric Transplantation (IMPACT) trial [15], children who received ATG induction therapy exhibited elevated frequencies of CD4^+^ Treg cells at both 1 and 3 months post-transplantation. In contrast to our study, the IMPACT trial demonstrated no significant difference in immunological risk between the ATG induction and control groups. Furthermore, regarding clinical outcomes, the ATG induction group displayed higher alloimmune event-free survival. Despite these differences, it is worth noting that our study results regarding Treg cell expression after ATG induction were consistent with those observed in the IMPACT trial. In our study, at 12 weeks post-ATG administration, CD4^+^CD25^+^CD127low T cell (representing Treg cells) expression was significantly increased compared with that at baseline, only in the ATG group. This finding further supports previous research suggesting a potential role for Treg cells in the preventive effects of ATG induction against acute rejection [23].

In the ATG group, CD8^+^CD28nullCD57^+^ T cell (representing senescent CD8^+^ T cells) expression tended to decrease at 4 weeks, followed by an increase at 12 weeks. In our previous study [14], CD8^+^CD28nullCD57^+^ T cells were highly expressed in patients with TCMR compared with normal controls. However, it is important to note that the time points in the aforementioned study differed from those in the current one because it was a cross-sectional study conducted on patients who had already experienced TCMR. CD8^+^CD28nullCD57^+^ T cells are generated through repeated antigenic stimulation during aging. These cells express high levels of adhesion and cytolytic molecules that enhance their cytotoxicity [24]. The decrease in CD8^+^CD28nullCD57^+^ T cell expression during the early post-transplantation period suggests a potent protective effect of ATG induction.

In the ATG group, a consistent decline in CD8^+^CCR7^+^ T cell expression was observed over time, and their frequency was consistently lower than that in the BXM group at all time points. Specifically, CD8^+^CCR7^+^CD45RA^-^ and CD8^+^CCR7^+^CD45RA^+^ T cell (representing CD8^+^ central memory T and CD8^+^ naïve T cells, respectively) expression significantly decreased at 12 weeks in the ATG group. These findings are consistent with those of previous studies [25,26], which reported that CD8^+^ naïve T cells remained at low levels, whereas effector CD8^+^ T cells remained elevated up to 5 years after transplantation. Furthermore, no differences in CD8^+^ naïve T and CD8^+^ effector memory T cells (CD8^+^CCR7^-^ T cells) were observed between patients with acute rejection at 6 months post transplantation and those with coronary artery disease (CAD) at 2 years post transplantation. However, decreased CD8^+^ naïve T cell and increased CD8^+^ effector memory T cell frequencies in patients with CAD were reported 5 years after transplantation [26]. These findings suggest that alterations in CD8^+^ T cell subsets may exhibit distinct dynamics depending on specific time points following transplantation.

Our previous research regarding CD8^+^CCR7^+^ T cell expression noted an increase in effector memory T cells and a decrease in CD8^+^ naïve T and CD8^+^ central memory T cells in patients with TCMR [12]. In the present study, we observed a decrease in naïve and central memory T cells and an increase in effector memory T cells in the ATG group, though TCMR occurrence was not significantly higher in this group. This observation does not consider the timing of rejection onset, and the limited observation during the early post-transplantation period could be a potential confounding factor. Further long-term research is necessary to verify these observations and confirm the absence of confounding influences during the early transplantation phase.

When analyzing patients diagnosed with BPAR, no significant changes were observed in other T cell subsets except for a significant decrease in CD4^+^ T cells. Although not statistically significant, an early post-transplantation increase in CD4^+^CD161^+^ T cells and a decrease in CD4^+^CD25^+^CD127low T cells were observed. Twelve weeks after transplantation, a decrease in CD8^+^CCR7^+^ T cells and an increase in CD8^+^CD28nullCD57^+^ T cells were observed, consistent with our previous research results [12,14]. In 8 of the 13 patients with BPAR (61.5%), CD8^+^CCR7^+^ T cell expression increased at 4 weeks compared with that at baseline. This finding suggests that CD8^+^CCR7^+^ T cell expression plays a key role in BPAR development.

There were some limitations to this study. First, as this study analyzed changes in lymphocyte subsets up to 12 weeks post transplantation, we could not evaluate the long-term effects of induction therapy on the immune system. Second, because the follow-up period was not long enough, clinical outcomes, such as chronic rejection or de novo DS-HLA generation, could not be observed. Third, due to the short follow-up period, there were few clinical events such as BPAR or allograft loss; therefore, it was difficult to detect meaningful differences in clinical outcomes according to induction therapy and changes in T cell subsets. Furthermore, given the absence of a validation cohort, it is essential to conduct future prospective cohort studies to validate the changes observed in T cell subsets.

Despite these limitations, our study possesses strengths in the serial assessment of immunological responses based on the type of induction therapy. Furthermore, we are conducting follow-up studies to validate if the immunological changes identified in our study are consistently observed and if they can help identify high-risk patients for rejection. Preliminary data from some patients can be found in Appendix A. This study is still in its early stages, and due to practical constraints, such as limited resources, we have only examined data from five patients who received ATG induction and eight patients who received BXM induction, at 3 months after KT. Baseline characteristics are provided in Appendix A. Similar to the current study population, the ATG group had more patients who were PRA-positive or DSA-positive, as well as a greater average HLA mismatch number when compared to the BXM group. When we looked at the levels of certain immune cells (CD4^+^CD25^+^CD127low T cells and CD4^+^CD161^+^ T cells), we noticed that there seemed to be higher levels in the ATG group compared to the BXM group, as shown in Appendix A. We have not examined other types of immune cells yet, and because our group of patients is small, we have not conducted any formal statistical analyses. Nevertheless, we have observed trends similar to our main study results. Additional research for validation purposes will further strengthen the quality of our findings.

## 4. Materials and Methods

### 4.1. Study Population

This retrospective observational study included 157 patients who underwent KT at Seoul St. Mary’s Hospital between May 2018 and November 2020. The patients were classified into two groups based on the type of induction therapy they received: ATG and BXM groups (Figure 4). Induction therapy was selected based on the immunological risk assessment of each patient. ATG induction was performed in highly sensitized patients who were defined as those with PRA > 50%, crossmatch positivity, or DS-HLA positivity (MFI > 3000). ATG was administered for five consecutive days from the day of transplantation. On the transplantation day, 1.5 mg/kg of ATG (Thymoglobulin; Genzyme Corporation, Cambridge, MA, USA) was administered intraoperatively before graft perfusion. From the day after transplantation, 1.25 mg/kg/day of ATG was administered intravenously. BXM induction was performed in patients with low immunological risk. BXM 20 mg (Simulect; Novartis Pharmaceutical Corp., Basel, Switzerland) was infused intravenously 4 h before and 4 days after transplantation.

To analyze the T cell phenotype using flow cytometry, we collected blood samples within 5 days before KT as a baseline blood sample, and additional blood samples were collected at 4 and 12 weeks after KT. All participants provided written informed consent in accordance with the Declaration of Helsinki. This study was approved by the Institutional Review Board at Seoul St. Mary’s Hospital (KC19TESI0043 on 15 May 2019).

### 4.2. Flow-Cytometric Analysis of Peripheral Blood Lymphocytes

Whole blood collected in sodium heparin tubes was divided into two 5 mL FACS tubes (Falcon Polystyrene Round-Bottom Tubes; BD Biosciences, Franklin Lakes, NJ, USA) at 100 µL per tube. For FACS tube 1, a cocktail of fluorescent antibodies comprising anti-CD3-V450 (560365; BD Biosciences, Franklin Lakes, NJ, USA), CD4-FITC (555346; BD Biosciences, Franklin Lakes, NJ, USA), CD8-APC-H7 (560179; BD Biosciences, Franklin Lakes, NJ, USA), CD28-PE-Cy7 (560684; BD Biosciences, Franklin Lakes, NJ, USA), CD161-APC (550968; BD Biosciences, Franklin Lakes, NJ, USA), and CD57-PE (560844; BD Biosciences, Franklin Lakes, NJ, USA) was prepared and added to the whole blood. Similarly, for FACS tube 2, a different cocktail of fluorescent antibodies comprising anti-CD3-V450 (560365; BD Biosciences, Franklin Lakes, NJ, USA), CD4-FITC (555346; BD Biosciences, Franklin Lakes, NJ, USA), CD8-APC-H7 (560179; BD Biosciences, Franklin Lakes, NJ, USA), CCR7-PE-Cy7 (557648; BD Biosciences, Franklin Lakes, NJ, USA), CD45RA-PerCP-Cy5.5 (563429; BD Biosciences, Franklin Lakes, NJ, USA), CD25-APC (555434; BD Biosciences, Franklin Lakes, NJ, USA), and CD127-BV510 (563086; BD Biosciences, Franklin Lakes, NJ, USA) was added to the whole blood. After incubation and centrifugation, the samples were analyzed using a Flow Cytometry system (LSRFortessaTM Flow Cytometer System; BD Biosciences, Franklin Lakes, NJ, USA) (Figure 5).

### 4.3. Clinical Parameters

Participant demographic (age, sex, donor type, and primary renal disease) and immunological parameters (number of HLA mismatches, PRA, and DS-HLA) were determined. A patient was diagnosed with BPAR when allograft biopsy confirmed a diagnosis of acute rejection based on the 2018 Banff classification [27]. In addition, CMV and BK infection episodes and allograft function were evaluated.

### 4.4. Statistical Analyses

All statistical analyses were performed using SPSS version 25.0 (SPSS Inc., Chicago, IL, USA). Continuous variables are presented as mean ± standard deviation, and categorical variables are presented as counts and percentages. The *t*-test was used to analyze continuous variables with a normal distribution, and the Mann–Whitney U test was used for non-normally distributed variables. The chi-square and Fisher’s exact tests were used for categorical variables. Statistical significance was set at *p* < 0.05.

## 5. Conclusions

In conclusion, our study demonstrates that ATG induction therapy induces more pronounced T lymphocyte changes than BXM induction, including an increase in CD4^+^CD161^+^ and CD4^+^CD25^+^CD127low T cells and an early decrease in CD8^+^CD28nullCD57^+^ and CD8^+^CCR7^+^ T cells. Some of these changes were consistent with those previously reported to be related to acute rejection. However, it is essential to acknowledge that the limited sample size and relatively short study duration constrained the depth of our analysis. Therefore, further investigations with larger cohorts and longer follow-up periods are required to validate and expand upon our findings. Future research should provide a more comprehensive understanding of the impacts of different induction therapies on the immune system and their implications for clinical outcomes.

## Figures and Tables

**Figure 1 ijms-24-14288-f001:**
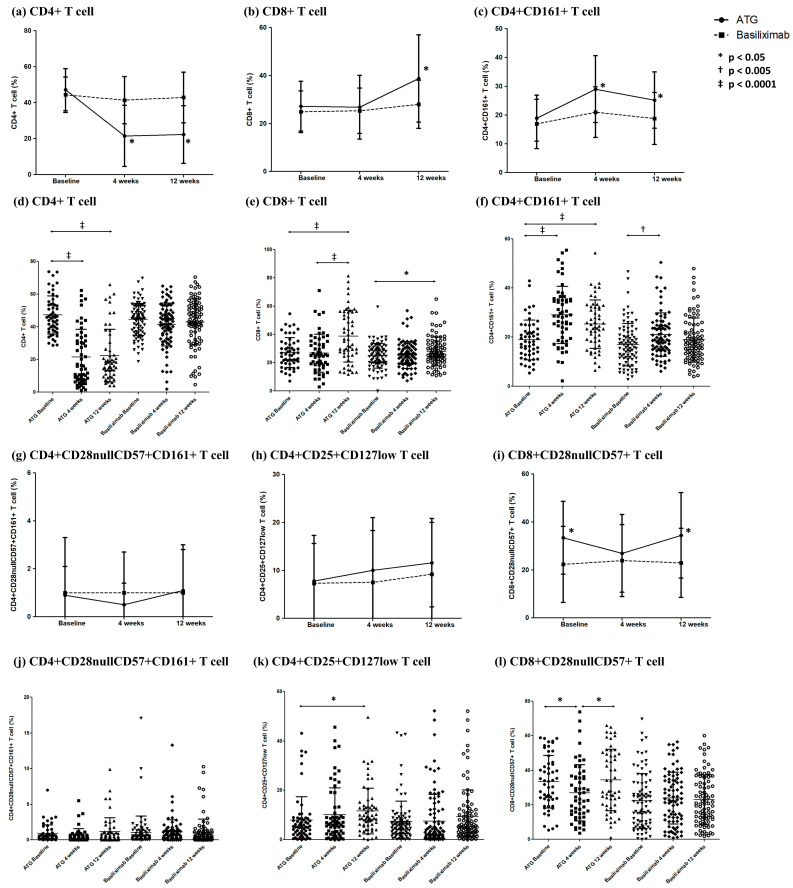
Changes in T cell subsets after transplantation. (**a**–**c**) The proportions of CD4^+^, CD8^+^, and CD4^+^CD161^+^ T cell subsets in the ATG and BXM groups. The solid line represents the average proportion of lymphocytes in the ATG group, and the dotted line represents that of the BXM group. (**d**–**f**) The serial lymphocyte subset proportion of patients in each group. (**g**–**i**) The proportions of CD4^+^CD28nullCD57^+^CD161^+^, CD4^+^CD25^+^CD127low, and CD8^+^CD28nullCD57^+^ T cell subsets in the ATG and BXM groups. (**j**–**l**) The serial lymphocyte subset proportion of patients in each group.

**Figure 2 ijms-24-14288-f002:**
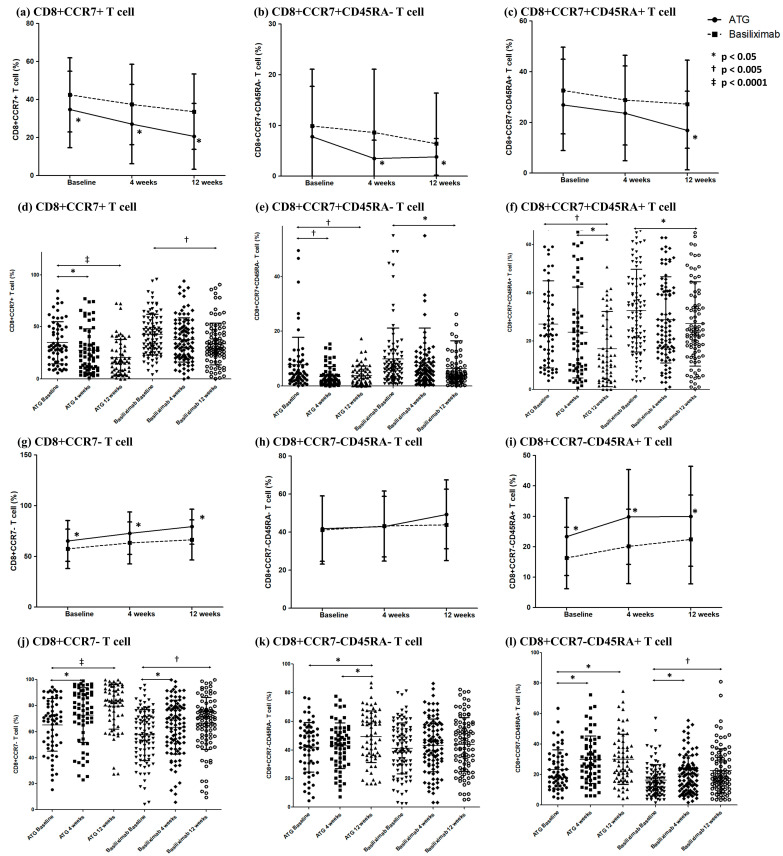
Changes in memory and effector CD8^+^ T cell subsets after transplantation. (**a**–**c**) The proportions of CD8^+^CCR7^+^, CD8^+^CCR7^+^CD45RA^-^, and CD8^+^CCR7^+^CD45RA^+^ T cell subsets in the ATG and BXM groups. The solid line represents the average proportion of lymphocytes in the ATG group, and the dotted line represents that of the BXM group. (**d**–**f**) The serial lymphocyte subset proportion of patients in each group. (**g**–**i**) The proportions of CD8^+^CCR7^-^, CD8^+^CCR7^-^CD45RA^-^, and CD8^+^CCR7^-^CD45RA^+^ T cell subsets in the ATG and BXM groups. (**j**–**l**) The serial lymphocyte subset proportion of patients in each group.

**Figure 3 ijms-24-14288-f003:**
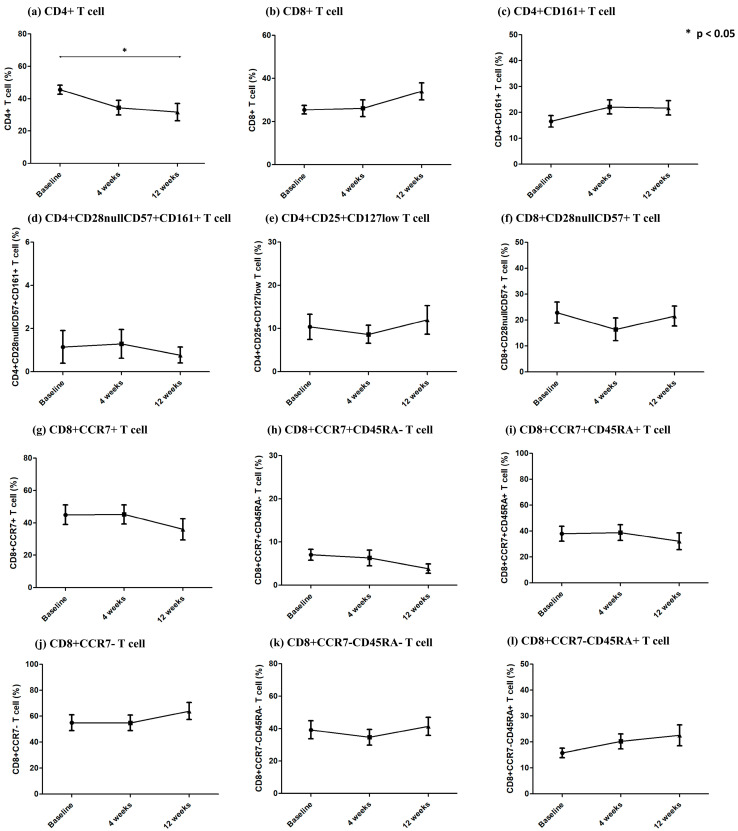
Changes in T cell subsets observed in patients with biopsy-proven acute rejection (BPAR). Among the 13 patients diagnosed with BPAR: (**a**) CD4^+^ T cell expression exhibited a significant decrease at 12 weeks. (**b**) CD8^+^ T cell expression increased at 12 weeks, although the increase was not statistically significant. (**c**–**f**) CD4^+^CD161^+^, CD4^+^CD28nullCD57^+^CD161^+^, CD4^+^CD25^+^CD127low, and CD8^+^CD28nullCD57^+^ T cell expression showed no significant changes over time. (**g**–**i**) There was no significant decrease was observed in CD8^+^CCR7^+^ T cell subsets. (**j**–**l**) No significant increase was observed in CD8^+^CCR7^-^ T cell subsets.

**Figure 4 ijms-24-14288-f004:**
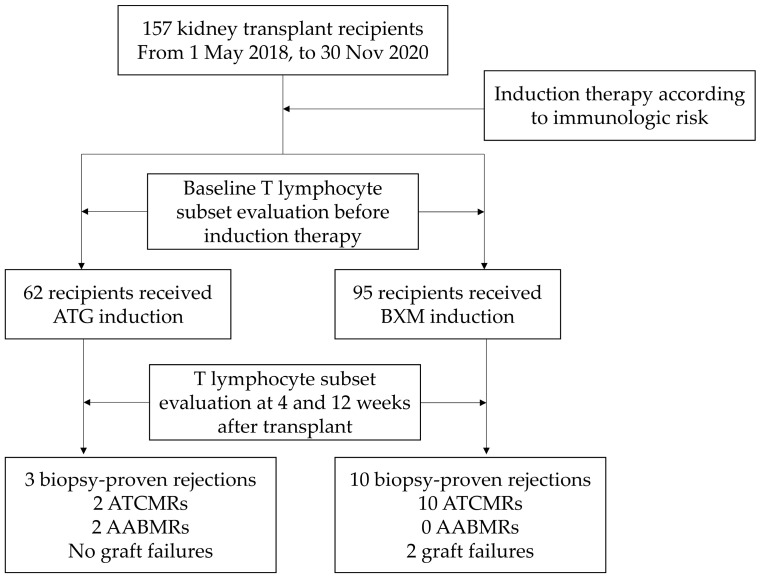
Scheme of the study.

**Figure 5 ijms-24-14288-f005:**
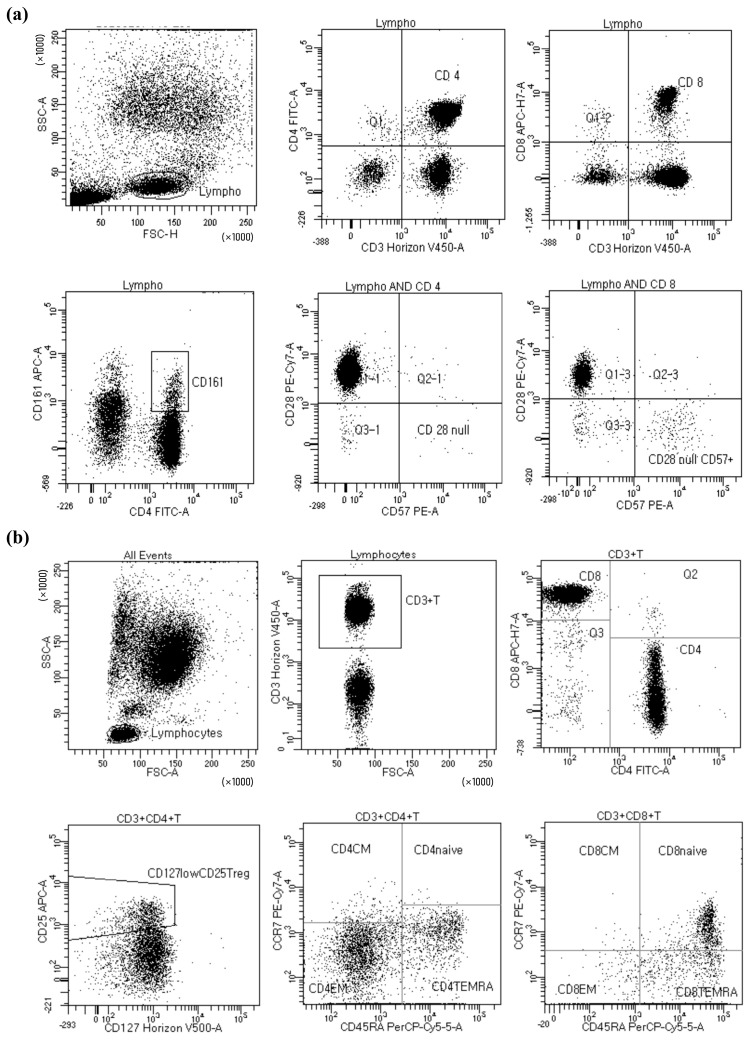
Flow cytometric analysis of T cell subsets. (**a**) Heparinized blood samples were stained with a cocktail of fluorescent antibodies targeting CD3, CD4, CD8, CD28, CD161, and CD57 to assess various T cell subsets, including CD4^+^, CD4^+^CD161^+^, CD4^+^CD28nullCD57^+^CD161^+^, CD4^+^CD25^+^CD127low, and CD8^+^, CD8^+^CD28nullCD57^+^ T cells. (**b**) A different cocktail of fluorescent antibodies targeting CD3, CD4, CD8, CCR7, CD45RA, CD25, and CD127 was used to assess CD8^+^CCR7^+^ and CD8^+^CCR7^-^ T cell subsets with or without CD45RA.

**Table 1 ijms-24-14288-t001:** Baseline characteristics.

	ATG (*n* = 62)	BXM (*n* = 95)	*p*-Value
Age (years)	50.1 ± 10.9	46.6 ± 11.7	0.061
Male, *n* (%)	34 (54.8)	59 (56.3)	0.408
Donor age (years)	48.1 ± 11.8	47.5 ± 11.5	0.748
Donor type (LD), *n* (%)	60 (96.8)	93 (97.9)	0.663
Primary disease, *n* (%)			0.053
DM	12 (19.4)	30 (31.6)	
HTN	9 (14.5)	12 (12.6)	
CGN	22 (35.5)	32 (33.7)	
PCKD	3 (4.8)	9 (9.5)	
Others	4 (6.5)	7 (7.4)	
Unknown	12 (19.4)	5 (5.3)	
HLA mismatch number	3.47 ± 1.65	3.15 ± 1.65	0.238
PRA-positive, *n* (%)	36 (58.1)	37 (38.9)	0.022
PRA Class I	29.1 ± 39.3	12.3 ± 26.5	0.002
PRA Class II	38.3 ± 43.9	12.0 ± 27.2	<0.001
DS-HLA antibody, *n* (%)	23 (37.1)	12 (12.6)	<0.001
Rituximab use, *n* (%)	58 (93.5)	28 (29.5)	<0.001

Data are expressed as means ± standard deviations or number (percentage). Abbreviations: ATG, anti-thymocyte globulin; CGN, chronic glomerulonephritis; PCKD, polycystic kidney disease; DS-HLA, donor-specific human leukocyte antigen; BXM, basiliximab.

**Table 2 ijms-24-14288-t002:** Clinical outcomes according to induction therapy.

	ATG (*n* = 62)	BXM (*n* = 95)	*p*-Value
BPAR, *n* (%)	3 (4.8)	10 (10.5)	0.206
ATCMR, *n* (%)	2 (3.2)	10 (10.5)	0.127
AABMR, *n* (%)	2 (3.2)	0 (0)	0.154
CMV infection, *n* (%)	0 (0)	0 (0)	N/A
BK viremia, *n* (%)	2 (3.2)	11 (11.6)	0.063
Graft failure, *n* (%)	0 (0)	2 (2.1)	0.519

Data are expressed as number (percentage). Chi-squared and Fisher’s exact tests were used to compare the incidence of clinical outcomes between the two groups. Abbreviations: BPAR, biopsy-proven acute rejection; ATCMR, acute T cell-mediated rejection; AABMR, acute antibody-mediated rejection; ATG, anti-thymocyte globulin; BXM, basiliximab.

**Table 3 ijms-24-14288-t003:** Comparison of T cell subsets according to induction therapy.

	ATG (*n* = 62)	BXM (*n* = 95)	*p*-Value
CD4^+^ T cell (%)			
Baseline	47.2 ± 11.7	44.4 ± 9.9	0.139
4 weeks after KT	21.5 ± 17.0	41.4 ± 13.1	<0.001
12 weeks after KT	22.3 ± 16.0	42.9 ± 14.1	<0.001
CD8^+^ T cell (%)			
Baseline	27.2 ± 10.4	24.9 ± 8.7	0.171
4 weeks after KT	26.8 ± 13.3	25.3 ± 9.5	0.496
12 weeks after KT	38.8 ± 18.2	28.0 ± 10.1	<0.001
CD4^+^CD161^+^ T cell (%)			
Baseline	18.9 ± 8.0	16.9 ± 8.6	0.167
4 weeks after KT	29.0 ± 11.6	21.0 ± 8.8	<0.001
12 weeks after KT	25.2 ± 9.8	18.8 ± 9.0	<0.001
CD4^+^CD28nullCD57^+^CD161^+^ T cell (%)			
Baseline	0.9 ± 1.2	1.0 ± 2.3	0.704
4 weeks after KT	0.5 ± 0.9	1.0 ± 1.7	0.054
12 weeks after KT	1.1 ± 1.9	1.0 ± 1.8	0.695
CD8^+^CD28nullCD57^+^ T cell (%)			
Baseline	33.4 ± 15.2	22.3 ± 15.9	<0.001
4 weeks after KT	26.9 ± 16.2	23.9 ± 15.0	0.267
12 weeks after KT	34.4 ± 17.8	22.9 ± 14.4	<0.001
CD4^+^CD25^+^CD127low T cell (%)			
Baseline	7.8 ± 9.5	7.3 ± 8.3	0.700
4 weeks after KT	10.0 ± 11.0	7.5 ± 10.8	0.158
12 weeks after KT	11.6 ± 9.2	9.2 ± 10.8	0.160
CD8^+^CCR7^+^ T cell (%)			
Baseline	34.8 ± 20.1	42.5 ± 19.5	0.018
4 weeks after KT	27.1 ± 20.9	37.4 ± 21.2	0.003
12 weeks after KT	20.6 ± 17.3	33.6 ± 19.8	<0.001
CD8^+^CCR7^+^CD45RA^+^ T cell (%)			
Baseline	26.9 ± 18.0	32.6 ± 17.1	0.051
4 weeks after KT	23.6 ± 18.7	28.8 ± 17.7	0.082
12 weeks after KT	16.8 ± 15.5	27.2 ± 17.4	<0.001
CD8^+^CCR7^+^CD45RA^-^ T cell (%)			
Baseline	7.8 ± 9.9	9.9 ± 11.2	0.234
4 weeks after KT	3.5 ± 3.6	8.6 ± 12.5	<0.001
12 weeks after KT	3.8 ± 3.6	6.4 ± 10.0	0.022
CD8^+^CCR7^-^ T cell (%)			
Baseline	65.2 ± 20.1	57.4 ± 19.5	0.018
4 weeks after KT	72.8 ± 20.9	63.3 ± 20.7	0.006
12 weeks after KT	79.3 ± 17.3	66.3 ± 19.8	<0.001
CD8^+^CCR7^-^CD45RA^+^ T cell (%)			
Baseline	23.3 ± 12.8	16.3 ± 10.1	<0.001
4 weeks after KT	29.8 ± 15.6	20.1 ± 12.2	<0.001
12 weeks after KT	30.0 ± 16.4	22.4 ± 14.6	0.003
CD8^+^CCR7^-^CD45RA^-^ T cell (%)			
Baseline	41.8 ± 17.2	41.1 ± 17.9	0.800
4 weeks after KT	42.9 ± 15.9	43.2 ± 18.4	0.927
12 weeks after KT	49.3 ± 18.1	43.8 ± 18.8	0.074

Data are expressed as means ± standard deviations. *t*- and Mann–Whitney U tests were used to compare the lymphocyte subsets of the two groups. Abbreviations: KT, kidney transplant; ATG, anti-thymocyte globulin; BXM, basiliximab.

## Data Availability

The data presented in this study are available in the article.

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
