# Peer review of "Impact of Induction Immunosuppressants on T Lymphocyte Subsets after Kidney Transplantation: A Prospective Observational Study with Focus on Anti-Thymocyte Globulin and Basiliximab Induction Therapies"

_ijms, 2023, doi:10.3390/ijms241814288_

Round 1

Reviewer 1 Report

Maintaining a balance between effective rejection prevention and the severity of immunosuppressive side effects is key to success in kidney transplantation. The authors presented an interesting study on “„Impact of Induction Immunosuppressants on T Lymphocyte Subsets After Kidney Transplantation: A Prospective Observational Study with Focus on Anti-thymocyte Globulin and Basiliximab Induction Therapies”. The authors have investigated the potential disparities in immune cell phenotypes based on different induction therapies to elucidate the differential impacts of these therapeutic approaches on clinical outcomes. The manuscript is clear, relevant for the field and presented in a well-structured manner. The experimental design is appropriate. In my opinion, the subject taken up by the authors is important and topical, but the manuscript needs improvement.

1.      Discussion. The authors conducted a discussion based on the results of their previous research. I propose to conduct the discussion also taking into account the results of other research teams.

2.      Line 273. Please complete the Bioethics Committee approval number with the date of approval.

3.      Please indicate the producers of the medicinal products used in the study (ATG, BXM).

4.      I suggest that the authors prepare a graphic scheme of the study.

Reviewer 2 Report

Kim et.al performed prospective observational study on induction immunosuppressive therapy for kidney transplant recipients, with a focus on the T cell subset characterization after BXM and ATG treatment. The entire manuscript focused on describing the difference of T cell subset with insufficient interpretation of the result. Below are my major concerns:

1. What is the hypothesis of this manuscript? Does T cell subsets change correlate anything related to kidney transplantation? Authors need to explain better about the significance of the study.

2. In result part (2.2), what is the conclusion from the clinical outcomes according to induction therapy? Were all rejections related to the change of T cell subset or they were just random cases?

3. Authors need to explain the biological/clinical meaning of each T cell subset and correlate them with the change observed in distinct treatment group at different time points.

4. Is there any validation cohort to support the T cell subsets change?

Round 2

Reviewer 1 Report

I recommend article for publication.

Author Response

We sincerely thank you for your thoughtful review and for recommending our article for publication. We greatly appreciate the time and effort you have dedicated to evaluating our work. Your feedback and constructive comments have been invaluable in improving the quality and clarity of our manuscript. We have carefully considered your suggestions and have made the necessary revisions accordingly. We believe that your insights have strengthened our research, and we are grateful for your input. Thank you for your continued support and consideration.

Reviewer 2 Report

The authors addressed most of my concern except the validation cohort, but I still believe that the validation (even partial of the cohort) should be included for publication. Therefore, I would like to see their ongoing effect (preliminary data) to support their points and strengthen their quality /novelty. It is encouraged to include them in the supplement so we know it is ongoing.
